# The Influence of Biofilm on Selected Properties of Thin-Coat Mineral-Based Plasters on EPS Substrate

**DOI:** 10.3390/ma15175963

**Published:** 2022-08-29

**Authors:** Monika Dybowska-Józefiak, Maria Wesołowska

**Affiliations:** Faculty of Civil and Environmental Engineering and Architecture, Bydgoszcz University of Science and Technology, 85-796 Bydgoszcz, Poland

**Keywords:** biocorrosion, ETICS system, facade surface

## Abstract

This paper discusses changes in the microstructure and water absorption of thin-coat mineral-based plasters after prolonged exposure to the external environment and infected with biological corrosion. The results of laboratory and field tests for external thermal insulation composite system Styrofoam-based plasters are presented. The test samples were taken after 6 years of exposure to the external environment. The microstructure parameters such as porosity distribution and water absorption of the plasters were evaluated. The pore size ranges that were sensitive to frost corrosion and capillary flow were separated in the porosity distribution. Based on the porosity and absorption changes, it was found that biological corrosion interferes with the microstructure of the thin-coat mineral-based plasters on the expanded polystyrene substrate.

## 1. Introduction

Beneficial long-term properties combined with excellent protection against blowing rain and high thermal insulation quality are the reasons why ETICSs (external thermal insulation composite systems) [1,2,3] have become so popular in Central Europe. The multilayer exterior coating is evaluated as a less vulnerable solution in comparison with traditional plaster coatings [4,5,6]. However, earlier ETICS applications revealed particularly poor resistance to mechanical damage and biological infestation of plasters [7]. Changes to the exterior wall’s finished surface due to microbial growth are generally accepted as ‘patina’ as long as they are uniformly distributed, while local contamination or algae concentrations are often evaluated as a ‘visually adverse’ [8] (Figure 1). However, the primary cause of plaster biocorrosion is microorganism presence.

Bacteria and fungi are involved in the degradation of any natural material. It has been found that from 20 to 30% of all damage is caused by microorganisms [5,9]. The process of biological deterioration of materials is referred to as biodeterioration, biocorrosion, or biological corrosion and is closely related to the organism metabolism [9]. Non-organic materials are exposed to corrosion with acidic metabolites produced by microorganisms (e.g., oxalic acid, gluconic acid, and citric acid) that form ion compositions with, e.g., lime and magnesium ions, taken from the material, which contributes to structure weakening [9,10]. Microorganisms are deposited onto surfaces to form clusters, surrounded by a protective layer of mucus, which are called biofilms or biological films (Figure 2a,b). Different biofilm areas are differentiated with density, organic matter availability, and oxygen content. Consequently, organisms in the film are characterized with different metabolic activity or even no activity at all [7].

Such behavior minimizes the diffusion of unfavorable substances from the outside (e.g., biocides) and enhances colony survival [10]. The biofilm affects changes in the appearance and structure of the engineering material [9]. It causes changes in porosity, is associated with vapor diffusion inside the material, absorbs contaminants, and leads to a change in the aerobic conditions into anaerobic conditions in the occupied area [7]. Acids, produced as a by-product of respiration and photosynthesis of biofilm-forming organisms, are often the cause of exterior wall finish deterioration [9]. This form of biodegradation is presented by bacteria, e.g., sulfur bacteria, that oxidize sulfur-containing nutrients to sulfuric acid and decompose scale or nitrifying bacteria with the production of corrosive nitric acid [10]. In case of building structures, the term biodeterioration can often be found, which means a general loss in the quality of building materials as a consequence of biological factors [10].

The study by Dylla et al. [11] on infected plaster surfaces of the ETICS system on selected structures in Bydgoszcz (Poland) found no bacteria and cyanobacteria that are often a component of aerophytic populations. However, three taxa of algae from the Chlorophyceae were successfully determined. By far, the dominant species was Apatococcus vulgaris Brand emend, Geitler—a cosmopolitan, aerophytic taxon with broad ecological valence. Filamentous green algae thrived in the wettest areas: Chlorohormidium flaccidum (Kütz.) Fott var. falaccidum Braun and the typically aquatic Ulothrix oscillarina (Kütz.) that is quite resistant to desiccation.

Changes in porosity and pore structure take place along with abiotic factors. According to a study conducted by Bochen, Gil, and Szwabowski, the average pore size decreases over time, but the open porosity volume increases [12]. In turn, water absorption and total porosity play key roles in microorganism growth on the surface of building materials, both of which are directly responsible for water and nutrient retention on and inside the samples [13].

Neville [14] and other researchers [8] believe that material structure tightness is essential for water and gas penetration. These parameters significantly affect strength, density, sorptivity, capillary suction, absorbability, permeability, and freeze–thaw durability. Therefore, they are used in methods for predicting the durability of building materials [12]. Water absorption and retention in materials are controlled by total porosity and pore size distribution (Tran at al. [15]). The effect of porosity and its structure on the physical properties of building materials was confirmed by Neville [14], Fagerlund [8], and other researchers [16,17,18,19,20,21,22], who conducted studies on concrete, mortar, cement paste, ceramic stone, gravel, and other materials. It is well known that a smaller pore volume has a good effect on the properties of cementitious materials since the tightness of material structure is of fundamental importance in terms of the penetration effected by water and gases. These parameters substantially influence the strength, density, sorptivity, capillary suction, absorption, permeability, and frost resistance, which are applied in the durability prediction methods of building materials [12,17,19,20,22]. The hardening process, which can take more than a year, can also increase the number of micropores and mesopores of less than 100 nm in size and reduce the number of capillary pores, which improves the material tightness. This promotes less penetration of moisture, and SO_2_ and NO_x_ aggressive gases [23] into the microstructure of plasters and thus improves the material durability. For this reason, the types and size of pores are quite important, especially open pores that allow for penetration of liquids and gases. According to Neville [14], a large number of pores less than 5 nm in size hinders water penetration. In pore sizes of 5–100 nm, water can flow due to diffusion. In pore sizes larger than 100 nm, water can flow due to capillary adsorption. In addition, at low temperatures in pores of 100–1000 nm in size, freezing water causes the largest damage in material [8]. With the expansion of areas containing capillary water, a more intense capillary flow occurs [24]. According to [25], capillary absorption and transport of water are only possible in pores in the range from 100 nm to 100,000 nm in size, which are also called capillary pores.

The aim of this study is to evaluate the changes in the microstructure of mineral thin-coat plasters after long-term exposure to environmental conditions and to determine the effect of biofilm on the degradation of these plasters.

## 2. Test Material

Test materials were mineral-based dashed plasters of the ETICS system. The study was performed on three sample groups:Initial laboratory samples (designated as group A);Initial laboratory samples after freeze–thaw cycles (designated as group B);Samples taken from the test bed subjected to a natural ageing process (designated as group C).

Samples for laboratory tests (Figure 3) were prepared according to ETICS system specification on a 2 cm thick Styrofoam insulation layer with dimensions of 75 × 300 mm that ensure the required surface area of 225 cm^2^. The study was performed on a group of 2 × 6 samples conditioned for 28 days in laboratory conditions.

The samples to be tested after freeze–thaw cycles were dried to a constant weight at +40 ± 2 °C. They were then soaked in water according to the procedure described in Section 3.1. The samples were placed in a climate chamber that was programmed with the following cycles:Heating up to 60 °C at 80% humidity within 1 h and maintaining constant temperature and humidity for 3 h;Reducing temperature to –20 °C within 1 h and maintaining it for 3 h.

After three complete cycles (24 h), the samples were weighed, and the humidity level of the samples was supplemented by soaking them in water according to the procedure described in Section 3.1 and weighed again. All 81 aging cycles were conducted this way. The number of cycles was derived from an analysis of temperature transitions through 0 °C during the period of test stand operation.

The test site was located on the campus of the University of Technology in Bydgoszcz, Poland. The site consists of four walls made of aerated concrete, 24 cm thick, peripherally insulated with 15 cm thick EPS70 Styrofoam. The surfaces of the plaster coatings were oriented on the test bench with respect to all directions in order to determine the influence of this factor as a determinant of the aesthetics of the plaster coatings. The protective function in ETICS systems is mainly performed by structural plaster. They should be characterized by the following properties: low absorbency, diffusivity, resistance to dirt, resistance to biological infestation, and UV resistance [14,15]. Therefore, four different layers of plaster were laid on each surface of the study area, separated by expansion joint profiles: acrylic, mineral, silicate, and silicone. The plasters were outdoor exposed for 6 years (2016–2022).

The test sample was obtained with the quartering method. To conduct the tests, samples of 10 cm × 10 cm were taken from the site, from the north and south elevations. Sampling began by cutting the plaster surface with an angle grinder and then using a knife to cut out the fragments along with the thermal insulation layer (Figure 4).

After sampling, a fragment of Styrofoam was cut off, leaving a 2 cm thick layer. The type C samples prepared this way were dried to a constant weight at +40 ± 2 °C.

The order of testing was based on the standard size of samples taken for testing, which were fragmented at a later stage. In the first stage, the following tests were performed: water absorption and freeze–thaw cycles for group B samples.

Within each group of samples, material was collected for microstructure determination. To ensure the proper representation of the results, material was taken from each of the 6 samples and then combined. The test sample was obtained with the quartering method.

## 3. Test Methods

### 3.1. Water Absorption Test

The water absorption for plasters was determined according to EN ISO 15148 [26]. Three samples were prepared for each plaster type, each with an area >100 cm2. The lateral surface of each sample was protected with silicone. The samples were then dried to a constant weight and placed in a water-filled vessel equipped with:A system to maintain a constant water level with an accuracy of ±2 mm;A load to hold the sample in a particular position;Spacers to hold the sample at least 5 mm from the bottom (Figure 5).

The first mass measurement was made after 5 min: the sample was taken out of the water, dried with a damp sponge, and then weighed with an accuracy of ±0.1% of its mass (Figure 6). The procedure of immersion, removal, drying, and weighing was repeated after 20 min, 1 h, 2 h, 4 h, 8 h, 12 h, and 24 h. For samples taken from the field site, the test was performed in two variants: with biofilm and after cleaning the plaster surface with biocide. The infected surface was pre-treated to remove bloom before the application of the biocide. The samples were evenly sprayed with biocide and left for 12 h. The contamination was then removed mechanically using high-pressure water. The preparation was applied twice with a 12 h interval.

The water absorption coefficient was determined based on the following formulas:(1)$$A_{w}=\frac{∆m_{tf}^{\prime }-∆m_{0}^{\prime }}{\sqrt{t_{f}}}$$
where:

$∆m_{tf}^{\prime }$—a value of ∆*m* read from the timeline $t_{f}$, in kg/m^2^;$∆m_{0}^{\prime }$—a value of ∆*m* read from the timeline $t_{f}$ for $t_{f}$ = 0, in kg/m^2^;$t_{f}$—test time, in seconds.

(2)$$W_{w}=\frac{∆m_{tf}^{\prime }-∆m_{0}^{\prime }}{\sqrt{t_{f}}}$$
where

$∆m_{tf}^{\prime }$—a value of ∆*m* read from the timeline $t_{f}$, in kg/m^2^;$∆m_{0}^{\prime }$—a value of ∆*m* read from the timeline $t_{f}$ for $t_{f}$ = 0, in kg/m^2^;$t_{f}$—test time, in hours.

### 3.2. Microstructure Test Using Mercury Intrusion Porosimetry (MIP)

The microstructure test was performed using a 9500 series AutoPore IV mercury intrusion porosimeter equipped with two ports: low and high pressures with a maximum value of 33,000 psi (228 MPa), which allows for measurements in the range of meso- and macropores. A penetrometer designed for granular and dusty material was used during test phase. Before conducting tests, the penetrometer was calibrated to determine the volume, compressibility, and thermal effect of the penetrometer used. An equilibrium time of 30 s was determined based on control measurements. As a result of the MIP test, the following parameters of the structure in question were determined: total pore volume, its skeletal density in mercury, and the distribution of the pore volume as a function of the pore diameter as an integral and differential relation.

The share of pore volume was calculated based on the following formulae:Pores responsible for freeze–thaw durability:
(3)$$U_{frost}=\frac{\sum_{i=100\;\mathrm{nm}}^{\;\;\;\;\;1000\;\mathrm{nm}}IV_{frost}}{TIV}\cdot P$$

Pores responsible for capillary transport:

(4)$$U_{cap}=\frac{\sum_{i=100\;\mathrm{nm}}^{\;\;\;\;\;100,000\;\mathrm{nm}}IV_{cap}}{TIV}\cdot P$$
where

*IV_frost_*—pore volume in the diameter range from 100 to 1000 nm;*IV_cap_*—pore volume in the diameter range from 100 to 100,000 nm;*TIV*—total mercury intrusion;*P*—total porosity.

## 4. Test Results

### 4.1. Water Absorption

Symptoms of biological corrosion occurred after two years, with varying degrees of severity depending on the orientation relative to the world directions (Figure 6).

The absorption results of tested samples are presented Table 1.

The water absorption coefficient A_w_ decreased twofold after the freezing cycles (type B samples) compared with the output samples (type A samples)—the value of A_w_ coefficient decreased from 0.0023 kg/(m^2^×s^0.5^) to 0.0011 kg/(m^2^×s^0.5^).

The field site samples where the biofilm layer is located show a lower water absorption coefficient than the initial samples A_w_ = 0.0023 kg/(m^2^×s^0.5^), with negligible differences between S and N orientation.

After sample cleaning and removing the biofilm as a result, the A_w_ index for the northern samples in S orientation is 0.0016 kg/(m^2^×s^0.5^) and that in the N orientation is 0.0036 kg/(m^2^×s^0.5^).

After biofilm removal, the southern orientation showed an increase in the A_w_ coefficient from 0.0007 kg/(m^2^×s^0.5^) to 0.0016 kg/(m^2^×s^0.5^)). In the northern orientation, however, the A_w_ coefficient increases from 0.0008 kg/(m^2^×s^0.5^) to 0.0036 kg/(m^2^×s^0.5^) (Table 1).

Large dynamic of water absorption in samples with biofilm removed can be observed in the N orientation from the beginning of the study (Figure 7); these changes are three times greater than the changes in the sample before cleaning. This relationship is not observed in the S orientation.

### 4.2. Microstructure

The microstructure results are shown in Figure 8 and Figure 9 and in Table 2 and Table 3.

Total porosity for the initial plaster samples was 34.2%. As the samples were subjected to specific freezing cycles, the porosity decreased to 26.8%. Field site samples show lower porosity than the initial samples. The porosity for the northern orientation is 29.2%, and the porosity for the southern orientation is 29.5%.

The volume share of pores related to capillary transport (from 100 nm to 100,000 nm) for the initial samples was 30.0%. As a result of freeze–thaw cycles in laboratory conditions, the value decreased to 21.2%. The utilization of plasters under environmental conditions resulted in a decrease in the value of the initial pore share for the S orientation to 24.8% and for the N orientation to 22.8%.

The pore share related to freeze–thaw durability (100 nm to 1000 nm) for the initial samples was 4.4%. After the freeze–thaw cycles in the laboratory, it increased to 8.8%. In case of the field site in the S orientation, it is 9.0% and in N orientation, it is 7.7%. Based on the analysis of the integral curves in the porosity range of 100–1000 nm (Figure 10), it was found that the pore volume in the dimension range of 300–1000 nm increased when compared with the initial sample.

When analyzing the mercury intrusion in mL/g for the specific pore ranges (Table 3), it can be concluded that the direct volume changes are very large. For the pore range related to capillary transport (from 100 nm to 100,000 nm), the largest changes were observed for laboratory freeze–thaw cycles—volume reduction of 36.8%—and the smallest for the south-oriented field site station—volume reduction of 20.8%. For the pore range related to freeze–thaw durability, similar results were obtained for freeze–thaw cycles and for the field site station in the N orientation—volume increases of more than 65%. Samples taken from the S-oriented field site station show almost twofold increase in mercury intrusion.

## 5. Discussion

Comparing the northern and southern orientations, it can be seen that the north-oriented plaster features higher water absorption than the south-oriented plaster. This relationship is visible in both non-cleaned samples as well as in samples with biofilm removed. 

On the other hand, when comparing field site samples with biofilm and samples subjected to freeze–thaw cycles, it can be observed that field site samples cut from the station have lower Aw coefficient than samples subjected to freeze–thaw cycles. On the other hand, after cleaning them, the Aw coefficient increases to 0.0016 kg/(m^2^×s^0.5^) in the case of the southern sample and to 0.0036 kg/(m^2^×s^0.5^) in the case of the northern sample, and at the same time, it is higher than the Aw coefficient for the samples tested in the climate chamber, where Aw = 0.00116 kg/(m^2^×s^0.5^).

Based on microscopic images, it was determined that the removal of biofilm from the N-oriented plaster compromises the plaster surface and texture (Figure 10).

The initial samples are featured by the lowest share of pores related to the freeze–thaw durability; these pores constitute 12.9% of total porosity. On the other hand, the initial samples show the highest share of pores related to capillary transport—87.7% of the total porosity. The dominant diameter of 3000 nm, determined from the differential pore size distribution, is within the range of capillary pores but outside the range of pores related to their freeze–thaw durability.

For the samples subjected to freeze–thaw cycles, the pore share in relation to total porosity for freeze–thaw durability is 30.6%, and the pore share for capillary transport is 79.1%. As a result of freeze–thaw cycles, the dominant diameter of 3000 nm disappeared, but a new dominant diameter of 1000 nm appeared in the pore region related to capillary transport.

Compared with the samples subjected to freeze–thaw cycles, the plaster samples taken from the field site station feature a higher pore share related to capillary transport: for the N orientation, by 78.1% and, for the S orientation, by 84.4%. On the other hand, the pores related to freeze–thaw durability for the field site represent 26.4% of the total porosity for the N orientation and 30.5% for the S orientation. The porosity distribution of south-oriented plaster is similar to the plaster subjected to freeze–thaw cycles. In case of the northern orientation, a bimodal porosity distribution takes place, where the dominant diameter of 3000 nm persists and, in addition, a second dominant diameter of 1000 nm is formed in the pore range related to capillary transport.

## 6. Conclusions

This paper presents the results of the study of thin-layer mineral-based plasters with a microorganism biofilm that colonizes exterior wall finishes. For comparison purposes, initial samples, samples subjected to laboratory freeze–thaw cycles, and samples from a south-oriented and north-oriented field test station were tested. On the field site station, the biofilm visible to the unaided eye appeared already after 2 years and developed for further 4 years. The number of freeze–thaw cycles in the laboratory test corresponded to the recorded number of passes through 0 °C over a 6-year period.

As a result of the study, changes in the absorption and the porosity structure of the thin-coat mineral-based plasters taken from the field site as well as in the samples subjected to freeze–thaw cycles were found. 

The test for water absorption changes showed a correlation with mercury intrusion and porosity changes in the capillary range in the case of cyclic freeze–thaw and a southern orientation of the field site station. For the northern orientation of the field site station, no such correlation was found—after biofilm removal with biocides, there are very high dynamics of water absorption during the test period (24 h) and, consequently, a very high coefficient of water absorption at similar porosity. This is probably the result of intensive biofilm growth and removal treatment procedures that disturb the surface and texture of the thin-coat mineral-based plaster.

The literature details the porosity range related to the freeze–thaw durability of plasters from 100 nm to 1000 nm. During the course of the study, it was found that both cyclic freeze–thaw and the utilization of plaster in a natural environment introduced changes in a narrower range of diameters, that is from 300 to 1000 nm. A common dominant diameter of 1000 nm also appeared in this range. Therefore, it is suggested that the porosity range responsible for the freeze–thaw durability of thin-coat mineral-based plasters should be narrowed from 300 nm to 1000 nm.

Further studies are expected in order to analyze the change in composition of the plasters due to freeze–thaw cycles and further development of biofilm on the plasters.

## Figures and Tables

**Figure 1 materials-15-05963-f001:**
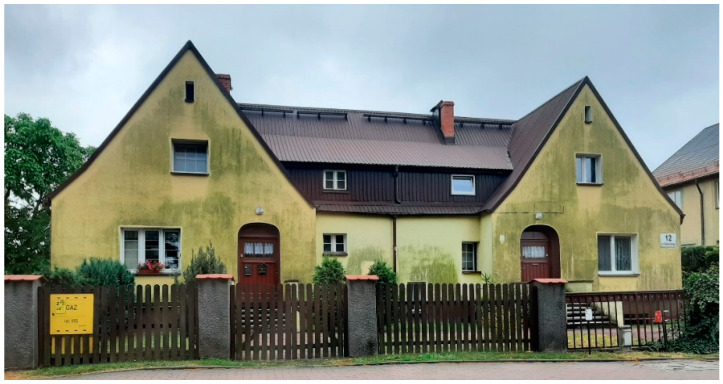
Changes in the exterior wall’s finished surface (author’s archives).

**Figure 2 materials-15-05963-f002:**
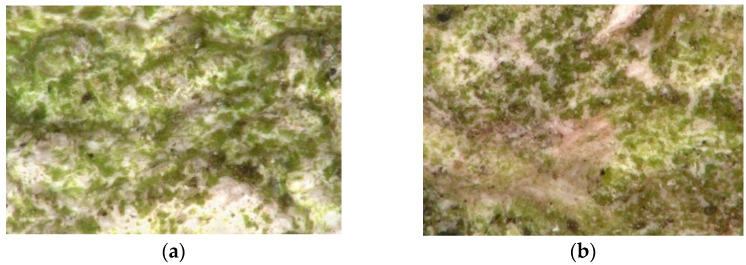
The surface of the (**a**) northern mineral-based dashed plaster—200× magnification photo taken with KEYENCE digital microscope (VHX-7000); (**b**) northern mineral-based filled plaster—200× magnification photo taken with KEYENCE digital microscope (VHX-7000).

**Figure 3 materials-15-05963-f003:**
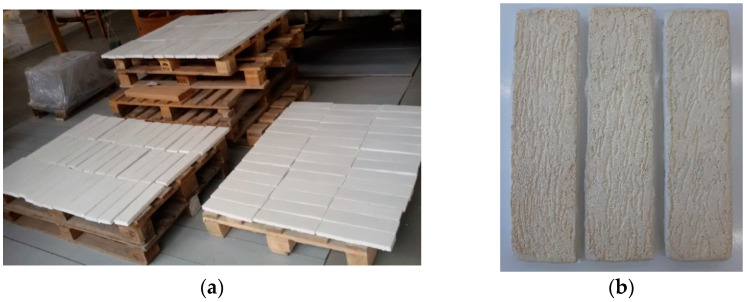
Plaster samples for laboratory tests: (**a**) sample conditioning method; (**b**) shape and sample texture (filled plaster).

**Figure 4 materials-15-05963-f004:**
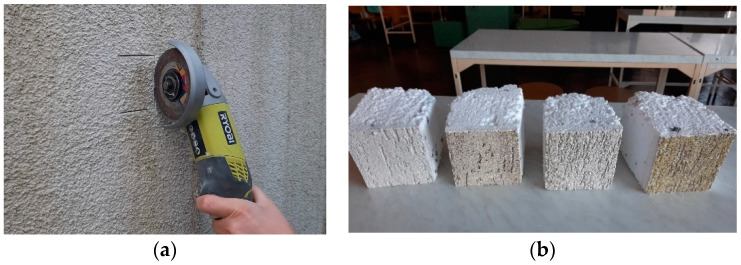
Sampling at the field site: (**a**) cutting the plaster surface; (**b**) samples that were cut out.

**Figure 5 materials-15-05963-f005:**
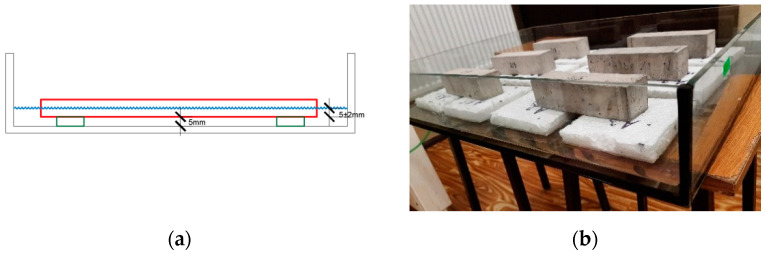
Test stand for tests of plaster water absorption: (**a**) schematic diagram; (**b**) view.

**Figure 6 materials-15-05963-f006:**
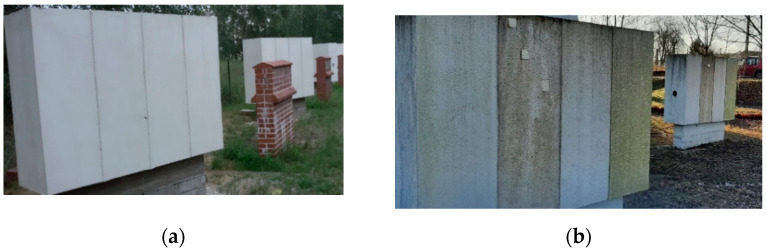
Test site for natural aging test: (**a**) 2016; (**b**) 2022.

**Figure 7 materials-15-05963-f007:**
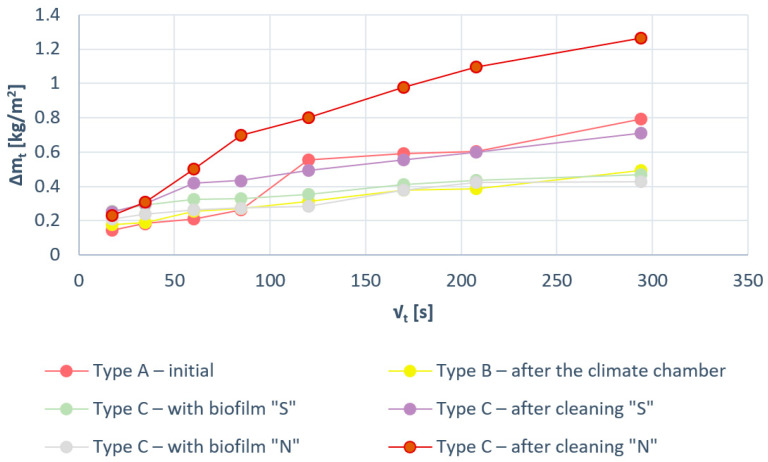
Graphs of ∆m_t_/F as a function of the time element t^0.5^ related to mineral-based plaster samples.

**Figure 8 materials-15-05963-f008:**
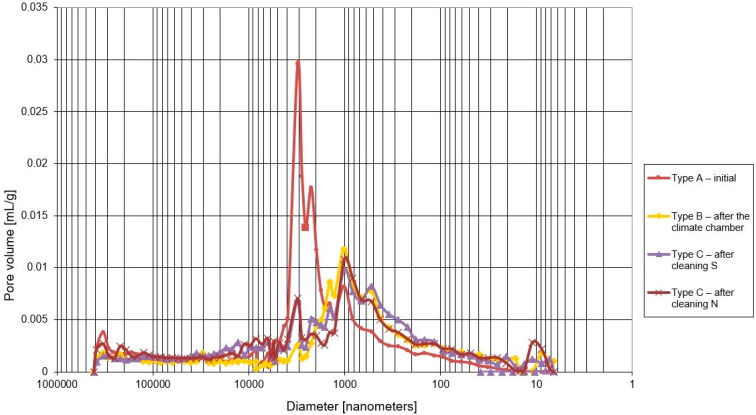
Differential curves of pore distribution for mineral-based plaster.

**Figure 9 materials-15-05963-f009:**
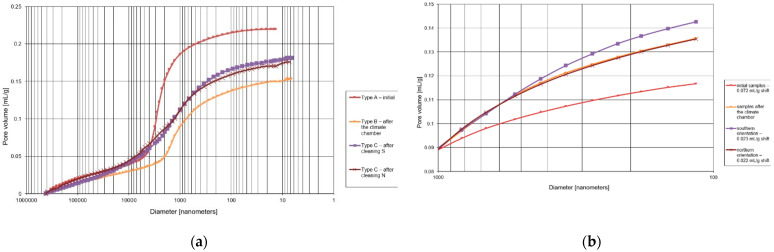
Integral curves of pore distribution for mineral-based plaster: (**a**) full measuring range; (**b**) pore range related to freeze–thaw durability (from 100 nm to 1000 nm).

**Figure 10 materials-15-05963-f010:**
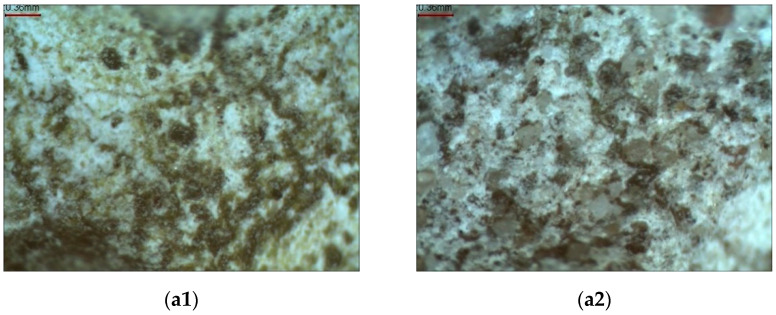
Photographs of the plaster surface with biofilm and after cleaning for the N and S orientation: (**a1**) N-oriented sample with biofilm; (**b1**) N-oriented sample after cleaning; (**a2**) S-oriented sample with biofilm; and (**b2**) S-oriented sample after cleaning.

**Table 1 materials-15-05963-t001:** Water absorption coefficient of mineral-based plaster.

Sample Type	Orientation	Water Absorption Coefficient	
		A_w_ (kg/(m^2^ × s^0.5^))	W_w_ (kg/(m^2^ × h^0.5^))
Initial (A)	-	0.0023	0.1372
After freeze–thaw cycles (B)	-	0.0011	0.0656
Taken from the field site (C) with biofilm	S	0.0007	0.0433
	N	0.0008	0.0466
Taken from the field site (C) after biocide cleaning	S	0.0016	0.0969
	N	0.0036	0.2177

**Table 2 materials-15-05963-t002:** Porosity test results.

Sample Type	General Porosity (%)	U Cap (%) (100 nm to 100,000 nm)	U Frost (%) (100 nm to 1000 nm)	Dominant Diameters (nm)
Initial (A)	34.2	30.0	4.4	3000
After freeze–thaw cycles (B)	26.8	21.2	8.2	1000
Taken from the field site (C), S orientation	29.5	24.8	9.0	1000
Taken from the field site (C), N orientation	29.2	22.8	7.7	1000 3000

**Table 3 materials-15-05963-t003:** Mercury intrusion for defined pore ranges related to capillary transport and frost corrosion.

Sample Type	Mercury Intrusion for a Given Range of Pores, mL/g		Volume Changes, mL/g	
	*IV* _ *cap* _	*IV* _ *frost* _	*IV* _ *cap* _	*IV* _ *frost* _
	100 nm ÷ 100,000 nm	100 nm ÷ 1000 nm	100 nm ÷ 100,000 nm	100 nm ÷ 1000 nm
Initial (A)	0.192562	0.02791	–	–
After freeze–thaw cycles (B)	0.121733	0.046944	36.8	−68.2
Taken from the field site (C), S orientation	0.152485	0.054965	20.8	−96.9
Taken from the field site (C), N orientation	0.137783	0.046643	28.4	−67.1

## Data Availability

Other results are not published.

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
