# Peer review of "The Influence of Biofilm on Selected Properties of Thin-Coat Mineral-Based Plasters on EPS Substrate"

_materials, 2022, doi:10.3390/ma15175963_

Round 1

Reviewer 1 Report

The manuscript focuses on the porosity changes and water absorption of thin-coat mineral based plaster due to environmental conditions. Some changes are suggested to improve the manuscript quality.  

  1. Please spell out acronyms in abstract.
  2. Line 23 “However, earlier ETICS applications revealed particularly poor resistance to impact and biological infestation of plasters” – this statement is not clear. What did the authors mean by impact?
  3. References should be re-formatted to correct style.
  4. Line 70 “It is well known that a smaller pore volume has a good effect on the properties of cementitious materials [13,15,16,18].” Which properties?
  5. Line 113 “The walls were oriented with respect to the directions of the world, in order to determine the influence of this factor as determining the aesthetics of the plaster coatings.” Please revise, repetition.
  6. Line 115 “Four different types of plasters …. “. It should be more specific what type of plaster, what are the differences between them, which types of plaster were selected for analysis?
  7. Lines 121 - 141, few sentences are repeated. Please delete.
  8. Line 123 “The order of tests was adapted to the grinding stages of the samples. In the first stage, the following tests were performed: water absorption and freeze-thaw cycles for group B samples.” This statement is not clear.
  9. Line 185, Table 1. Details of biocide cleaning should be mentioned in methods.
  10. Line 188, “The water absorption coefficient Aw decreased twofold after the freezing cycles (type B samples) compared to the output samples (type B samples)”. Did the authors mean compared to initial samples, type A?
  11. Figure 8, Figure labels should be replotted and in correct format.
  12. Line 296, recommendation section needs revising.

Author Response

Dear Reviewer,

Thank you very much for your valuable tips related to the article.

Reviewer 2 Report

The paper entitled ‘The influence of biofilm on selected properties of thin-coat mineral-based plasters on EPS substrate’ intends to correlate the influence of biofilm on the water absorption and porosity of ETICS with EPS as thermal insulation layer. Although some infos can be interesting for researchers and expert in this field, a major review is necessary prior to publication. The Materials and Methods section is rather confusing, and a discussion (with proper correlation among data and literature) is completely missing. No clue on the biofilm composition, or on the biodeterioration due to outdoor exposure, was provided. Furthermore, few points should de addressed:

-What do you mean with ‘samples taken from the test stand (designated as group C)’? are these reference specimens?

- Caption of fig. 3: include it in the caption ‘Plaster samples for laboratory tests: a) Sample conditioning method (b ) Shape and sample … ; please do the same also for the following pictures.

- This part:

The samples to be tested after freeze-thaw cycles were dried to a constant weight at +40±2°C. They were then soaked in water according to the procedure described in section 3.1. The samples were placed in a climate chamber that was programmed with the following cycles:

• heating up to 60°C at 80% humidity within 1 hour and maintaining constant temperature and humidity for 3 hours;

• reducing temperature to –20°C within 1 hour and maintaining it for 3 hours.

After 3 complete cycles (24 hours), the samples were weighed, and the humidity level of the samples was supplemented by soaking them in water according to the procedure de scribed in section 3.1, and weighed again. All 81 aging cycles were conducted this way. The number of cycles was derived from an analysis of temperature transitions through 109

0°C during the period of test stand operation.’

should be in the method section, not in the material section. Please review accordingly.

- Concerning freeze-thaw cycles, did you follow any norm? Please refer it accordingly, was it any adapted or modified norm?

- ‘The walls were oriented with respect to the directions of the world’: what do you mean? It was oriented in the four geographical directions or just north/south? Please correct and specify;

- ‘The plasters prepared this way operated in the environment’: rather use ‘the plasters were outdoor exposed for..’

- ‘Symptoms of biological corrosion occurred after two years, with varying degrees of severity depending on the orientation relative to the world directions (Figure 4).’: this part should be moved in the result section

- ‘The order of tests was adapted to the grinding stages of the samples.’: what does it means?

- This section is definitely confusing; it should be reviewed. The last 7 sentences of this section were even repeated;

Results:

- Figure 7: picture have rather low luminosity, especially 1b and 2b, please correct it accordingly;

- A discussion section is missing: this section is supposed to comprehensively correlate your data, otherwise the paper can be considered only a report and a comparison among data. I would expect also some hypothesis based on the obtained data, using also references of other authors to compare your data. This latter part is totally missing, for instance there is no mention on the biodeterioration of the plasters (as widely argued in the introduction);

- Furthermore> what about the biofilm mentioned in the beginning of the paper? What is the biofilm supposed to be? Fungi, algae, cyanobacteria or something else? Further details should be provided;

- The conclusions should be only a brief summary of the obtained data;

- ‘This section is not mandatory but can be added to the manuscript if the discussion is unusually long or complex’: what is the sense of this sentence at the end of the conclusion?

Author Response

(The authors gave the same response as above.)

Reviewer 3 Report

Research on the durability of materials exposed to aggressive climatic conditions has high importance and practical applicability. The authors have developed a complex and well-conceptualized experimental work.

Introduction is detailed and provides useful information, well supported by the cited references. A minus is that the bibliographic documentation studied is quite old; I recommend the authors to complete the documentary study with some more recent articles.

Research methodology is described accurately, being supported by eloquent images, of good quality. In my opinion it would be more appropriate for sub-chapters 2 and 3 to be included in a single chapter entitled "Materials and methods"; it is just a suggestion. There are some minor superscript and subscript typos (lines 74, 96).

Experimental results are presented and analyzed accordingly, the graphic images are able to support the discussions.

Conclusions are quite detailed. I recommend that they be reduced to the main findings of the research, but some of the details be included in the Discussions. I appreciate that the authors refer to future studies that plan to be carried out for the in-depth analysis of the structural changes of the material.

Author Response

(The authors gave the same response as above.)

Round 2

Reviewer 2 Report

Significant improvements were introduced in the manuscript, in accordance with the revision. Although a further connection among literature might be provided in the discussion (no references were included), the paper can be accepted for publication